# Modifiable and non-modifiable risk factors for COVID-19, and comparison to risk factors for influenza and pneumonia: results from a UK Biobank prospective cohort study

Frederick K Ho ,[1] Carlos A Celis-Morales,[1,2] Stuart R Gray ,[2]
S Vittal Katikireddi ,[1] Claire L Niedzwiedz ,[1] Claire Hastie,[1] Lyn D Ferguson,[2]
Colin Berry,[2] Daniel F Mackay,[1] Jason MR Gill,[2] Jill P Pell,[1] Naveed Sattar ,[2]
Paul Welsh[2]

► Additional material is published online only. To view please visit the journal Online (http://dx.doi.org/10.1136/bmjopen-2020-040402).

[1]Institute of Health and Wellbeing, University of Glasgow, Glasgow, UK
[2]Institute of Cardiovascular and Medical Sciences, University of Glasgow, Glasgow, UK

**Correspondence to**
Dr Paul Welsh;
Paul.Welsh@glasgow.ac.uk

## ABSTRACT

**Objectives** We aimed to investigate demographic, lifestyle, socioeconomic and clinical risk factors for COVID-19, and compared them to risk factors for pneumonia and influenza in UK Biobank.

**Design** Cohort study.

**Setting** UK Biobank.

**Participants** 49–83 year olds (in 2020) from a general population study.

**Main outcome measures** Confirmed COVID-19 infection (positive SARS-CoV-2 test). Incident influenza and pneumonia were obtained from primary care data. Poisson regression was used to study the association of exposure variables with outcomes.

**Results** Among 235 928 participants, 397 had confirmed COVID-19. After multivariable adjustment, modifiable risk factors were higher body mass index and higher glycated haemoglobin (HbA1C) (RR 1.28 and RR 1.14 per SD increase, respectively), smoking (RR 1.39), slow walking pace as a proxy for physical fitness (RR 1.53), and use of blood pressure medications as a proxy for hypertension (RR 1.33). Higher forced expiratory volume in 1 s (FEV1) and high-density lipoprotein (HDL) cholesterol were both associated with lower risk (RR 0.84 and RR 0.83 per SD increase, respectively). Non-modifiable risk factors included male sex (RR 1.72), black ethnicity (RR 2.00), socioeconomic deprivation (RR 1.17 per SD increase in Townsend Index), and high cystatin C (RR 1.13 per SD increase). The risk factors overlapped with pneumonia somewhat, less so for influenza. The associations with modifiable risk factors were generally stronger for COVID-19, than pneumonia or influenza.

**Conclusion** These findings suggest that modification of lifestyle may help to reduce the risk of COVID-19 and could be a useful adjunct to other interventions, such as social distancing and shielding of high risk.

## INTRODUCTION

COVID-19, caused by SARS-CoV-2 infection, includes a spectrum of morbidity from

### Strengths and limitations of this study

► Large cohort from a general population, as opposed to a cohort of hospitalised patients, at an age relevant to more severe COVID-19 symptoms.
► Biochemistry assays performed in a single dedicated central laboratory.
► Extensively emerging and novel risk factors for COVID-19.
► UK Biobank is not representative of the whole UK population.
► Exposures were measured several years before the development of the outcomes, and misclassification of the exposures will likely bias our results to the null.

asymptomatic infection[1] to severe pneumonia in patients presenting for medical care.[2] The COVID-19 pandemic[3] has led to concerted research efforts to identify people at greatest risk of developing the infection and progressing to critical illness and dying. Predictors of disease severity include older age, smoking, diabetes, hypertension, kidney disease, chronic obstructive pulmonary disease (COPD) and previous cardiovascular disease (CVD).[4–7] In addition, it is becoming clear that other risk factors might include obesity and low physical fitness.[8 9] However, many studies investigate risk factors for disease progression to death or critical illness among hospitalised patients[7] as opposed to healthy comparators.

Identifying the risk factors for COVID-19 is important in terms of identifying factors that can be modified to reduce risk, as well as identifying non-modifiable risk factors that can help identify high risk groups who require

shielding and targeting for testing, and eventual vaccine and anti-viral therapies. Pneumonia is a life-threatening complication of COVID-19 infection.[10] Established major risk factors for community-acquired pneumonia include many of the emerging risk factors for COVID-19.[11] As COVID-19 is caused by SARS-CoV-2 viral infection, it may have risk factors for contraction of the disease that are common to other respiratory virus conditions such as Influenza.

UK Biobank is a large prospective, deeply phenotyped, population-based cohort study carried out in the UK.[12] Over the study period, testing for COVID-19 in England was conducted in accident and emergency (A&E) departments and in-hospital. These data were provided by Public Health England (PHE) and linked to UK Biobank baseline data. We aimed to establish modifiable and non-modifiable risk factors for confirmed COVID-19. We also aimed to compare these risk factors to risk factors for the incident pneumonia over a similar time span.

## METHODS
### Participants
UK Biobank was conducted via 22 assessment centres across England, Scotland and Wales between March 2006 and December 2010 and recruited 502 624 participants aged 37–73 years. The present study was restricted to participants living in England for whom COVID-19 test results were available. Death data were available to the end of January 2018 in England. To reduce competing risks, whereby risk factors may influence risk of death before the pandemic occurred, we excluded from the study all participants known to have died prior to the COVID-19 pandemic. Baseline biological measurements were recorded and touch-screen questionnaires were administered according to a standardised protocol.[12 13]

### Outcomes
Results of COVID-19 PCR tests for UK Biobank participants were provided by PHE.[7 14] Data provided by PHE included the specimen date, specimen type (eg, upper respiratory tract), laboratory, origin (whether evidence from microbiological record that the was conducted in hospital setting or not) and result (positive or negative). At the time the manuscript was developed,[15] data were available for the period 16 March 2020 to 3 May 2020. Results were available for 5356 tests conducted on 3003 individuals. Confirmed COVID-19 infection (primary outcome) was defined as at least one positive result in the context of an in-hospital or A&E test. Any positive test result was used as a sensitivity analysis. Longer follow-up of the UK Biobank cohort has become available, but in this analysis, we have elected to investigate risk factors for more severe disease (in-hospital positive tests) in the early stages of the pandemic.

Pneumonia was defined based on the 10th revision of the International Classification of Diseases, (ICD-10) codes J12–J18, and influenza based on J09–J11, converted into Read Codes using the UK Biobank's look-up table. Incident pneumonia and influenza, occurring after 1 January 2016 (an arbitrary date taken to mimic the time lag from baseline exposure measurement to incident COVID-19 infection, while also obtaining sufficient case numbers) was obtained from a 41% sample of participants with available data from primary care. A sensitivity analysis was conducted using cases after 1 January 2015 and demonstrated consistent data. An analysis was also conducted exploring risk factors for all cases of pneumonia and influenza after baseline.

### Exposures
Exposures were measured at the baseline assessment visit between 2006 and 2010. 'Modifiable' risk factors were considered to include smoking, anthropometric measurements, glycated haemoglobin (HbA1C), lung function measurements, hypertension, high-density lipoprotein (HDL) cholesterol, other lipid measurements, and measures of physical activity.

Current age (on 1 March 2020) was derived from the assessment date and age at recruitment. Ethnicity, smoking, alcohol consumption, physician-diagnosed prevalent conditions and medication use were self-reported. For the present analyses, ethnicity was coded as white, south Asian, black, or mixed/other. Smoking status was categorised into never vs former/current smoking. Systolic and diastolic blood pressures were measured at the baseline visit, preferentially using an automated measurement, but using manual measurement where this was not available, and average of available measures used. Area-level socioeconomic deprivation was assessed by the Townsend score (incorporating measures of unemployment, non-car ownership, non-home ownership and household overcrowding) corresponding to the participants' home postcode. Higher scores on the Townsend score represent greater socioeconomic deprivation. Self-reported walking pace was rated by each participant as slow, steady/average, or brisk.[16]

The definition of baseline diabetes included self-reported type 1 or type 2 diabetes, those with a primary or secondary hospital diagnoses relating to diabetes at baseline (ICD-10 codes E10-E14.9), and those who reported using diabetes medications. Baseline CVD was defined as self-reported myocardial infarction, stroke, or transient ischaemic attack. Cancer, longstanding illness, and depression were self-reported on touchscreen questionnaire. Other previous health complaints including asthma, rheumatoid arthritis, chronic kidney disease, systemic lupus erythematosus, sleep apnoea, COPD, pneumonia, bronchitis (including bronchitis, bronchiectasis and emphysema), and other respiratory diseases (including interstitial lung disease, asbestosis, pulmonary fibrosis, alveolitis, respiratory failure, pleurisy, pneumothorax, other respiratory condition) which were derived from self-report at nurse interview. Some conditions were not included in multivariable analysis due to low case numbers and/or high correlations with other conditions,

but univariable results are presented for completeness. Statin (categorised to include other cholesterol lowering medications) and blood pressure medication use were also recorded from self-report, with blood pressure medication being used as a proxy for baseline diagnosed hypertension.

Body mass index (BMI) is the ratio of the measured body mass in kg divided by height squared measured in metres. Height was measured using a Seca 202 height measure. Weight and whole-body fat mass and fat free mass were measured to the nearest 0.1kg using the Tanita BC-418 MA body composition analyser. Socks and shoes were removed when height was measured Grip strength was measured using a Jamar J00105 hydraulic hand dynamometer and the mean was derived from the right and left hand values expressed in kilograms.[17]

Lung function was assessed by spirometry using a Vitalograph Pneumotrac 6800 spirometer (Vitalograph, Buckingham, UK). Participants did not perform spirometry if they answered yes to unsure to the following: chest infection in the last month, history of detached retina, heart attack, surgery to eyes, chest or abdomen in last 3months, history of collapsed lung, pregnancy, or currently on medication for tuberculosis. The aim was to record two acceptable blows from a maximum of three attempts. The spirometer software compared the acceptability of the first two blows and, if acceptable (defined as a ≤5% difference in forced vital capacity (FVC) and forced expiratory volume in 1 s (FEV1)), the third blow was not required. The mean observation was taken for both measures.

Blood collection sampling procedures for the study have been previously described and validated.[18] Biochemical assays were performed at a dedicated central laboratory on around 480 000 samples. Further details of these measurements can be found in the UK Biobank Data Showcase and Protocol (http://www.ukbiobank.ac.uk). For the present study we included total cholesterol, HDL cholesterol, rheumatoid factor, cystatin C, HbA1C, C reactive protein (CRP), differential white cell count and red cell distribution width as exposures of interest. Biomarkers with data below the limit of detection were imputed as the square root of the limit of detection. The majority of participants had undetectable rheumatoid factor and risk ratios were derived for detectable rheumatoid factor, with the referent being undetectable.

## Statistics

Mean and SD were reported for continuous outcomes except for biomarkers, where median and IQR were reported. Poisson regression with robust 'sandwich' standard errors were used to study the associations of exposure variables with confirmed COVID-19 and pneumonia. Poisson regressions were used because they provide risk ratio (RR) which is easier to interpret and robust error estimation ensures accurate inference.[19] Three adjustment schemes were considered: model 0—univariate (ie, no adjustment), model 1—adjusted for age, sex, ethnicity and deprivation index, and model 2—further adjusted for

behavioural (smoking and alcohol drinking) and physical (adiposity, blood pressure, spirometry and physical capability) factors that were found to be significant in model 1. In model 2, there were variables that were derived from the same variable (eg, BMI and BMI categories) and have strong correlations (r=0.87 between BMI, and body fat mass, and r=0.97 between FEV1 and FVC). To avoid multicollinearity, in model 2, we chose one from, BMI, BMI categories, body fat-free mass, body fat mass and body fat per cent. and one from FEV1, FVC and FEV1/FVC. For continuous variables, the linearity of exposure-outcome associations were tested using penalised cubic splines in generalised additive model.[20] Nonlinearity was tested using likelihood ratio test comparing a model with the exposure fitted on a spline with a model assuming a linear exposure-outcome relationship. P value for nonlinearity <0.05 suggest evidence against the linearity assumption. Spline smoothness was chosen using generalised cross validation.[21] Population attributable fractions (PAFs) were calculated to determine the relative contribution of each risk factor to the overall number of confirmed COVID-19 cases within UK Biobank. Another Poisson model which included all significant factors in model 2 was fitted to estimate the mutually adjusted risk ratios (RRs). These RRs were biased towards null because of over adjustment bias but were used to construct the PAFs to ensure the PAF estimates did not overlap or exceed 100%. In general, two-tailed p values<0.05 were considered statistically significant. Analyses were conducted in R Statistical Software V.3.5.3 with the package 'mgcv'.

## Patient and public involvement

UK Biobank maintains a website and twitter feed to keep participants, the general public and researchers up to date on the study (http://www.ukbiobank.ac.uk/news/). There is an annual scientific meeting which is recorded and available to the public as webcast. The results of the present study are shared through these channels as the UK Biobank organisation deem appropriate, our own twitter feeds and open-access publication.

The study was set up by the MRC, Department of Health and Wellcome Trust with input from major patient representative organisations (British Heart Foundation and Cancer Research UK: http://www.ukbiobank.ac.uk/public-consultation/).

## RESULTS

Of 445 857 participants in England, 428 225 were alive during the available follow-up period. Complete data on covariates were available for 235 928 (55.1%) participants. Primary care data on incident pneumonia were available in 96 814 (41.0%) participants (online supplemental figure 1). At 1 March 2020, the age range of eligible participants was 49–83 years, and time elapsed from baseline was median 10.97 years (IQR 10.36–11.55 years). Of these participants, 1525 received at least one COVID-19 test, and 518 had confirmed SARS-CoV-2

infection, with 397 positive results conducted in hospital or A&E (primary outcome).

## Univariable risk factors for incident COVID-19

Key univariable potentially modifiable risk factors for confirmed COVID-19 included current and former smoking (RR 1.56), higher BMI, body fat and HbA1C (RR 2.32 for obesity), poor lung function (RR 0.84 for 1 SD increase in FEV1), treated hypertension (RR 1.89), HDL cholesterol (RR 0.71 per SD increase) and slow walking pace (RR 2.29; table 1). Among non-modifiable risk factors were older age (particularly the over 75 year olds; RR 1.44), male sex (RR 1.38), black ethnicity (RR 2.88), socioeconomic deprivation (RR 1.30 per SD increase in Townsend Index; table 1). Among baseline comorbidities, general long standing illness (RR 1.59), baseline diabetes (RR 2.16), baseline CVD (RR 1.93), sleep apnoea (RR 3.32), statin use (RR 1.80), higher inflammatory markers (particularly white blood cell count; RR 1.17 per SD increase) and higher cystatin C (RR 1.34 per SD increase) were also associated with confirmed COVID-19 (table 1). Risk factor associations were similar when all 518 test positive COVID-19 cases were considered, rather than only in-hospital positive tests (online supplemental table 1).

## Univariable risk factors for incident pneumonia

Of the 96 814 participants with primary care data, 209 had pneumonia recorded after 2016.

Of the modifiable risk factors, pneumonia was associated with BMI, body fat and HbA1C (more moderately than COVID-19; RR 1.43 for obesity), poor lung function (more strongly than COVID-19; RR 0.63 for 1 SD increase in FEV1), and slow walking pace (RR 2.07), but not smoking, lipids, blood pressure or blood pressure medication use. Among non-modifiable risk factors, incident pneumonia was more common in participants who were older (RR 2.09 in the over 75 year olds), and in women (RR 0.59 among men) (table 1). Ethnicity was not associated with pneumonia. Among comorbidities, baseline diabetes, CVD and cancer were approximately twice as common in those who developed pneumonia (table 1), and any longstanding illness was also associated with pneumonia. Inflammatory markers were similarly associated with pneumonia as with COVID-19.

When investigating risk factors for pneumonia over the full follow-up time from baseline, risk factors were generally similar although there was increased power due to a higher number of incident cases (online supplemental table 1). In this analysis, smoking was associated with pneumonia (RR 1.15) as was blood pressure medication use (RR 1.13). Similar findings were found when we studied pneumonia occurring after 2015 (online supplemental table 22).

## Univariable risk factors for incident influenza

Of the 96 814 participants with primary care data, 94 had influenza recorded after 2016.

Those who developed influenza were generally slightly more socioeconomically deprived (RR 1.20 per SD increase in Townsend score), more likely to have bronchitis at baseline, and less likely to take statins. No other risk factors showed associations (table 1). When all influenza cases occurring after baseline were considered, evidence of statistically significant associations emerged due to increased power. Smoking (RR 1.18), poor lung function (RR 0.84 for 1 SD increase in FEV), slow walking pace (RR1.35), and high BMI (RR 1.20 for obesity) were all associated with higher influenza risk (online supplemental table 1). Cases were less common in older people (RR 0.69 in the over 75 year olds) and more common in women (RR 0.83 in men). South Asians (RR 2.20) were at increased risk.

## Multivariable models for COVID-19 and pneumonia

After multivariable adjustment for age, sex, ethnicity and socioeconomic status, the modifiable risk factors for confirmed COVID-19 included smoking (RR 1.45 (95% CI 1.19 to 1.79)), higher BMI (RR 1.36 per SD increase (95% CI 1.25 to 1.48)) and other measures of body fat, higher HbA1C (RR 1.23 per 1 SD increase (95% CI 1.15 to 1.32)), as well as blood pressure medication, FEV1 and FVC, and slow walking pace (RR 1.99 (95% CI 1.48 to 2.68); model 1, table 2). Other risk factors were older age (RR 1.12 per 5 years (95% CI 1.05 to 1.19)), male sex (RR 1.36 (95% CI 1.11 to 1.66)), black ethnicity (RR 2.32 (95% CI 1.33 to 4.04)), South Asian ethnicity (RR 1.98 (95% CI 1.10 to 3.55)) socioeconomic deprivation (RR 1.27 per SD increase in Townsend Index (95% CI 1.16 to 1.39)). Among comorbidities, risk factors included long-standing illness (RR 1.44 (95% CI 1.17 to 1.77)), diabetes, CVD and statin use. Among blood biomarkers, risk factors included lower total and HDL cholesterol, higher cystatin C (RR 1.27 per 1 SD increase (95% CI 1.16 to 1.39)), CRP and white cell count (table 2).

After adding BMI, blood pressure, FEV1 and walking pace as mediators/covariates to the adjustment model, modifiable risk factors for COVID-19 admission continued to include smoking (RR 1.39 (95% CI 1.13 to 1.73)), higher BMI (RR 1.28 (95% CI 1.16 to 1.40)), higher HbA1C (RR 1.14 (95% CI 1.05 to 1.23), treated hypertension (RR 1.33 95% CI 1.04 to 1.70) and slow walking pace (RR 1.53 (95% CI 1.12 to 2.08)). The age association was no longer significant after these additional adjustments. Other risk factors included male sex (RR 1.72 (95% CI 1.25 to 2.35)), black ethnicity (RR 2.00 (95% CI 1.16 to 3.53)), socioeconomic deprivation (RR 1.17 (95% CI 1.06 to 1.29)), lower HDL cholesterol (RR 0.83 (95% CI 0.73 to 0.95)) and higher cystatin C (RR 1.13 (95% CI 1.02 to 1.25); model 2, table 2).

Nonlinear associations of continuous variables with COVID-19 and pneumonia are shown in figure 1 (,and online supplemental table 3). Age was associated with COVID-19 admission in a J-shaped curve, where for participants aged 60–70 years the curve was relatively flat, and for participants aged over 75 years risk increased

**Table 1** Univariable association of baseline risk factors with in-hospital COVID-19 in 2020, pneumonia occurring after 2016 and influenza occurring after 2016

| | In-hospital COVID-19 | | | | Pneumonia (after 2016) | | | | Influenza (after 2016) | | | |
|---|---|---|---|---|---|---|---|---|---|---|---|---|
| | No n=235531 | Yes n=397 | RR* | P value | No n=96605 | Yes n=209 | RR | P value | No n=96720 | Yes n=94 | RR | P value |
| Age (years)† | 66.53 (8.09) | 67.55 (8.68) | 1.14 | 0.01 | 66.52 (8.06) | 68.78 (7.60) | 1.34 | <0.0001 | 66.53 (8.06) | 66.20 (7.83) | 0.96 | 0.69 |
| Age categories (years), n (%) | | | | | | | | | | | | |
| <60 | 58683 (24.92) | 102 (25.69) | 1 | REF | 23707 (24.54) | 31 (14.83) | 1 | REF | 23717 (24.52) | 21 (22.34) | 1 | REF |
| 60–64 | 37589 (15.96) | 42 (10.58) | 0.64 | 0.02 | 15230 (15.77) | 24 (11.48) | 1.20 | 0.49 | 15233 (15.75) | 21 (22.34) | 1.56 | 0.15 |
| 65–69 | 42812 (18.18) | 58 (14.61) | 0.78 | 0.13 | 17606 (18.22) | 45 (21.53) | 1.95 | 0.004 | 17632 (18.23) | 19 (20.21) | 1.22 | 0.54 |
| 70–74 | 55726 (23.66) | 93 (23.43) | 0.96 | 0.78 | 23233 (24.05) | 63 (30.14) | 2.07 | 0.0009 | 23278 (24.07) | 18 (19.15) | 0.87 | 0.67 |
| ≥75 | 40721 (17.29) | 102 (25.69) | 1.44 | 0.009 | 16829 (17.42) | 46 (22.01) | 2.09 | 0.002 | 16860 (17.43) | 15 (15.96) | 1.00 | 0.99 |
| Male sex, n (%) | 111095 (47.17) | 219 (55.16) | 1.38 | 0.001 | 45548 (47.15) | 72 (34.45) | 0.59 | 0.0003 | 45573 (47.12) | 47 (50.00) | 1.12 | 0.58 |
| Ethnicity (%) | | | | | | | | | | | | |
| White | 224215 (95.20) | 356 (89.67) | 1 | REF | 92477 (95.73) | 201 (96.17) | 1 | REF | 92588 (95.73) | 90 (95.74) | 1 | REF |
| Black | 3057 (1.30) | 14 (3.53) | 2.88 | 0.0001 | 908 (0.94) | 3 (1.44) | 1.52 | 0.47 | 911 (0.94) | 0 (0.00) | 0.00 | 0.97 |
| South Asian | 3508 (1.49) | 12 (3.02) | 2.15 | 0.009 | 1605 (1.66) | 3 (1.44) | 0.86 | 0.80 | 1607 (1.66) | 1 (1.06) | 0.64 | 0.66 |
| Others | 4751 (2.02) | 15 (3.78) | 1.99 | 0.009 | 1615 (1.67) | 2 (0.96) | 0.57 | 0.43 | 1614 (1.67) | 3 (3.19) | 1.91 | 0.27 |
| Deprivation index (score) | −1.44 (2.98) | −0.56 (3.37) | 1.30 | <0.0001 | −1.48 (2.90) | −1.28 (3.07) | 1.07 | 0.31 | −1.48 (2.90) | −0.92 (3.32) | 1.20 | 0.06 |
| Current/former smoker, n (%) | 106829 (45.36) | 224 (56.42) | 1.56 | <0.0001 | 43843 (45.38) | 93 (44.50) | 0.96 | 0.80 | 43891 (45.38) | 45 (47.87) | 1.11 | 0.63 |
| Alcohol drinking status, n (%) | | | | | | | | | | | | |
| Never | 8894 (3.78) | 27 (6.80) | 1 | REF | 3714 (3.84) | 13 (6.22) | 1 | REF | 3723 (3.85) | 4 (4.26) | 1 | REF |
| Former | 7469 (3.17) | 19 (4.79) | 0.84 | 0.56 | 3090 (3.20) | 13 (6.22) | 1.20 | 0.64 | 3099 (3.20) | 4 (4.26) | 1.20 | 0.80 |
| Current | 219168 (93.05) | 351 (88.41) | 0.53 | 0.001 | 89801 (92.96) | 183 (87.56) | 0.58 | 0.06 | 89898 (92.95) | 86 (91.49) | 0.89 | 0.82 |
| BMI (kg/m²) | 27.32 (4.56) | 29.12 (5.25) | 1.40 | <0.0001 | 27.41 (4.56) | 28.61 (5.58) | 1.26 | 0.0002 | 27.42 (4.57) | 27.53 (4.70) | 1.03 | 0.80 |
| BMI categories, n (%) | | | | | | | | | | | | |
| Underweight | 787 (0.33) | 1 (0.25) | 1.17 | 0.88 | 305 (0.32) | 1 (0.48) | 1.65 | 0.62 | 306 (0.32) | 0 (0.00) | 0.00 | 0.97 |
| Normal | 78073 (33.15) | 85 (21.41) | 1 | REF | 31182 (32.28) | 62 (29.67) | 1 | REF | 31209 (32.27) | 35 (37.23) | 1 | REF |
| Overweight | 101751 (43.20) | 172 (43.32) | 1.55 | 0.0009 | 41867 (43.34) | 80 (38.28) | 0.96 | 0.81 | 41917 (43.34) | 30 (31.91) | 0.64 | 0.07 |
| Obese | 54920 (23.32) | 139 (35.01) | 2.32 | <0.0001 | 23251 (24.07) | 66 (31.58) | 1.43 | 0.04 | 23288 (24.08) | 29 (30.85) | 1.11 | 0.68 |
| Body fat-free mass (Kg) | 53.76 (11.47) | 56.18 (11.50) | 1.23 | <0.0001 | 53.75 (11.49) | 52.08 (12.04) | 0.86 | 0.04 | 53.74 (11.49) | 54.61 (12.17) | 1.08 | 0.46 |
| Body fat mass (Kg) | 24.49 (9.19) | 27.44 (10.35) | 1.32 | <0.0001 | 24.69 (9.21) | 27.90 (11.25) | 1.35 | <0.0001 | 24.70 (9.22) | 24.31 (9.37) | 0.96 | 0.68 |
| Body fat proportion (per cent) | 31.01 (8.40) | 32.36 (8.47) | 1.17 | 0.001 | 31.19 (8.41) | 34.29 (8.68) | 1.44 | <0.0001 | 31.19 (8.41) | 30.51 (8.46) | 0.92 | 0.43 |
| Systolic blood pressure (mm Hg) | 137.24 (18.07) | 138.49 (18.76) | 1.07 | 0.17 | 137.55 (18.10) | 137.55 (18.03) | 1.00 | 1.00 | 137.55 (18.10) | 137.60 (17.40) | 1.00 | 0.98 |
| Diastolic blood pressure (mm Hg) | 82.08 (9.90) | 82.99 (10.47) | 1.10 | 0.07 | 82.19 (9.90) | 80.90 (9.63) | 0.88 | 0.06 | 82.19 (9.90) | 82.79 (10.74) | 1.06 | 0.56 |
| FEV1 (L) | 2.86 (0.77) | 2.73 (0.78) | 0.84 | 0.0009 | 2.86 (0.78) | 2.53 (0.81) | 0.63 | <0.0001 | 2.86 (0.78) | 2.84 (0.88) | 0.98 | 0.84 |
| FVC (L) | 3.79 (0.98) | 3.68 (0.97) | 0.89 | 0.03 | 3.79 (0.99) | 3.40 (1.01) | 0.65 | <0.0001 | 3.79 (0.99) | 3.73 (1.04) | 0.95 | 0.60 |
| FEV1/FVC | 0.76 (0.06) | 0.74 (0.08) | 0.85 | 0.0004 | 0.76 (0.06) | 0.74 (0.08) | 0.82 | 0.0007 | 0.76 (0.06) | 0.76 (0.06) | 1.04 | 0.72 |
| Walking pace (%) | | | | | | | | | | | | |

Continued

**Table 1** Continued

| | In-hospital COVID-19 | | | | Pneumonia (after 2016) | | | | Influenza (after 2016) | | | |
|---|---|---|---|---|---|---|---|---|---|---|---|---|
| | No n=**235531** | Yes n=397 | RR* | P value | No n=**96605** | Yes n=209 | RR | P value | No n=**96720** | Yes n=94 | RR | P value |
| Slow | 14887 (6.32) | 59 (14.86) | 2.29 | <0.0001 | 6190 (6.41) | 29 (13.88) | 2.07 | 0.0005 | 6209 (6.42) | 10 (10.64) | 1.78 | 0.10 |
| Average | 123588 (52.47) | 213 (53.65) | 1 | REF | 50839 (52.63) | 115 (55.02) | 1 | REF | 50908 (52.63) | 46 (48.94) | 1 | REF |
| Brisk | 97056 (41.21) | 125 (31.49) | 0.75 | 0.010 | 39576 (40.97) | 65 (31.10) | 0.73 | 0.04 | 39603 (40.95) | 38 (40.43) | 1.06 | 0.78 |
| Grip strength (kg) † | 31.31 (10.93) | 31.36 (10.31) | 1.00 | 0.93 | 31.21 (10.96) | 27.83 (10.83) | 0.72 | <0.0001 | 31.20 (10.96) | 31.94 (11.27) | 1.07 | 0.51 |
| **Prevalent disease at baseline (%)** | | | | | | | | | | | | |
| Longstanding illness, n (%) | 68208 (28.96) | 156 (39.29) | 1.59 | <0.0001 | 28899 (29.91) | 89 (42.58) | 1.74 | <0.0001 | 28954 (29.94) | 34 (36.17) | 1.33 | 0.19 |
| Diabetes | 10969 (4.66) | 38 (9.57) | 2.16 | <0.0001 | 4575 (4.74) | 20 (9.57) | 2.12 | 0.001 | 4591 (4.75) | 4 (4.26) | 0.89 | 0.82 |
| CVD | 11238 (4.77) | 35 (8.82) | 1.93 | 0.0002 | 4870 (5.04) | 18 (8.61) | 1.77 | 0.02 | 4884 (5.05) | 4 (4.26) | 0.84 | 0.73 |
| Cancer | 15849 (6.75) | 32 (8.12) | 1.22 | 0.28 | 6557 (6.81) | 27 (12.92) | 2.03 | 0.0006 | 6573 (6.82) | 11 (11.70) | 1.81 | 0.06 |
| Depression | 45781 (19.44) | 68 (17.13) | 0.86 | 0.25 | 18651 (19.31) | 34 (16.27) | 0.81 | 0.27 | 18667 (19.30) | 18 (19.15) | 0.99 | 0.97 |
| CKD stages 3-5 | 293 (0.12) | 2 (0.50) | 4.04 | 0.049 | 128 (0.13) | 0 (0.00) | 0.00 | 0.97 | 128 (0.13) | 0 (0.00) | 0.00 | 0.98 |
| SLE | 274 (0.12) | 0 (0.00) | 0.00 | 0.96 | 125 (0.13) | 0 (0.00) | 0.00 | 0.97 | 125 (0.13) | 0 (0.00) | 0.00 | 0.98 |
| Asthma | 27722 (11.77) | 54 (13.60) | 1.18 | 0.26 | 11433 (11.83) | 34 (16.27) | 1.45 | 0.049 | 11454 (11.84) | 13 (13.83) | 1.19 | 0.55 |
| Sleep apnoea | 718 (0.30) | 4 (1.01) | 3.32 | 0.02 | 317 (0.33) | 0 (0.00) | 0.00 | 0.97 | 317 (0.33) | 0 (0.00) | 0.00 | 0.97 |
| COPD | 536 (0.23) | 3 (0.76) | 3.33 | 0.04 | 241 (0.25) | 0 (0.00) | 0.00 | 0.96 | 240 (0.25) | 1 (1.06) | 4.31 | 0.15 |
| Bronchitis | 2515 (1.07) | 8 (2.02) | 1.90 | 0.07 | 1085 (1.12) | 6 (2.87) | 2.59 | 0.02 | 1087 (1.12) | 4 (4.26) | 3.90 | 0.008 |
| Pneumonia | 2938 (1.25) | 8 (2.02) | 1.63 | 0.17 | 1152 (1.19) | 4 (1.91) | 1.61 | 0.34 | 1155 (1.19) | 1 (1.06) | 0.89 | 0.91 |
| Other respiratory disease | 1243 (0.53) | 3 (0.76) | 1.43 | 0.53 | 511 (0.53) | 1 (0.48) | 0.90 | 0.92 | 512 (0.53) | 0 (0.00) | 0.00 | 0.98 |
| **Medication at baseline, n (%)** | | | | | | | | | | | | |
| Statin | 36314 (15.42) | 98 (24.69) | 1.80 | <0.0001 | 15358 (15.90) | 39 (18.66) | 1.21 | 0.28 | 15389 (15.91) | 8 (8.51) | 0.49 | 0.055 |
| BP medication | 37590 (15.96) | 105 (26.45) | 1.89 | <0.0001 | 15719 (16.27) | 34 (16.27) | 1.00 | 1.00 | 15741 (16.27) | 12 (12.77) | 0.75 | 0.36 |
| Steroid | 876 (0.37) | 1 (0.25) | 0.68 | 0.70 | 354 (0.37) | 1 (0.48) | 1.31 | 0.79 | 355 (0.37) | 0 (0.00) | 0.00 | 0.98 |
| **Biomarker at baseline** | | | | | | | | | | | | |
| Total cholesterol (mmol/L) † | 5.62 (4.92-6.34) | 5.48 (4.68-6.21) | 0.85 | 0.001 | 5.62 (4.91-6.34) | 5.62 (4.84-6.30) | 0.99 | 0.94 | 5.62 (4.91-6.34) | 5.62 (5.04-6.37) | 1.07 | 0.49 |
| HDL cholesterol (mmol/L) | 1.40 (1.17-1.67) | 1.29 (1.09-1.54) | 0.71 | <0.0001 | 1.39 (1.17-1.67) | 1.41 (1.14-1.70) | 1.01 | 0.87 | 1.39 (1.17-1.67) | 1.36 (1.12-1.64) | 0.92 | 0.44 |
| Cystatin C (mg/L) | 0.88 (0.80-0.97) | 0.92 (0.83-1.02) | 1.34 | <0.0001 | 0.88 (0.80-0.97) | 0.89 (0.80-1.01) | 1.12 | 0.07 | 0.88 (0.80-0.97) | 0.91 (0.82-0.98) | 1.03 | 0.76 |
| HbA1c (mmol/mol) † | 35.10 (32.60-37.60) | 36.20 (33.60-39.60) | 1.30 | <0.0001 | 35.10 (32.60-37.60) | 36.00 (33.50-39.00) | 1.22 | <0.0001 | 35.10 (32.60-37.60) | 35.10 (32.32-38.30) | 1.08 | 0.39 |
| CRP (mg/L) † | 1.25 (0.63-2.56) | 1.58 (0.86-3.28) | 1.15 | 0.0001 | 1.27 (0.64-2.59) | 1.37 (0.84-3.27) | 1.14 | 0.01 | 1.27 (0.64-2.59) | 1.34 (0.70-2.93) | 1.07 | 0.47 |
| Rheumatoid factor (IU/ml) † | 3.16 (3.16-3.16) | 3.16 (3.16-3.16) | 1.02 | 0.64 | 3.16 (3.16-3.16) | 3.16 (3.16-3.16) | 1.04 | 0.50 | 3.16 (3.16-3.16) | 3.16 (3.16-3.16) | 1.04 | 0.64 |
| Red cell distribution width (%) | 13.30 (12.90-13.80) | 13.32 (12.97-13.95) | 1.12 | 0.01 | 13.30 (12.89-13.80) | 13.38 (12.97-14.02) | 1.15 | 0.02 | 13.30 (12.89-13.80) | 13.27 (12.97-13.70) | 0.90 | 0.36 |
| White cell count (*10^9/L) | 6.60 (5.61-7.80) | 6.90 (5.84-8.26) | 1.17 | 0.001 | 6.63 (5.64-7.80) | 6.90 (5.80-8.29) | 1.15 | 0.03 | 6.63 (5.64-7.80) | 6.60 (5.50-7.76) | 1.02 | 0.82 |
| Neutrophil count (*10^9/L) | 4.00 (3.25-4.90) | 4.12 (3.29-5.13) | 1.12 | 0.02 | 4.01 (3.26-4.92) | 4.11 (3.41-5.15) | 1.10 | 0.13 | 4.01 (3.26-4.92) | 4.02 (3.30-5.05) | 0.99 | 0.93 |
| Lymphocyte count (*10^9/L) | 1.88 (1.52-2.29) | 1.90 (1.55-2.40) | 1.13 | 0.009 | 1.88 (1.52-2.30) | 1.92 (1.60-2.36) | 1.11 | 0.12 | 1.88 (1.52-2.29) | 1.85 (1.51-2.53) | 1.04 | 0.68 |
| Monocyte count (*10^9/L) | 0.45 (0.36-0.56) | 0.47 (0.39-0.59) | 1.13 | 0.01 | 0.45 (0.37-0.57) | 0.48 (0.37-0.59) | 1.07 | 0.31 | 0.45 (0.37-0.57) | 0.50 (0.40-0.57) | 1.14 | 0.17 |

Continued

**Table 1** Continued

| | In-hospital COVID-19 | | | | Pneumonia (after 2016) | | | | Influenza (after 2016) | | | |
|---|---|---|---|---|---|---|---|---|---|---|---|---|
| | No n=235 531 | Yes n=397 | RR* | P value | No n=96 605 | Yes n=209 | RR | P value | No n=96 720 | Yes n=94 | RR | P value |

Numbers represent number (%) for categorical variables, mean (SD) for Gaussian continuous variables, and median (25–75 percentiles) for skewed variables.

*Univariable risk ratio per 1 SD for continuous variables and using comparator group for categorical variables.

†Evidence for non-linear association with COVID-19 (see online supplemental table 33)

BMI, body mass index; BP, blood pressure; CKD, chronic kidney disease; COPD, chronic obstructive pulmonary disease; CVD, cardiovascular disease; FEV1, forced expiratory volume in 1 s; FVC, forced vital capacity; SLE, systemic lupus erythematosus.

exponentially with age. The associations of BMI and FEV1 with COVID-19 were fairly linear. Associations were similar when adjusting for body fat percentage rather than BMI (online supplemental table 4).

Directly contrasting model 2 for COVID-19 and pneumonia (figure 2), the risk factors in common for both conditions were higher BMI and slow walking pace, although less strongly associated with pneumonia than for COVID-19. Highlighting specific differences, smoking and treated blood pressure were only associated with COVID-19. Pneumonia was more common in women, and socioeconomic deprivation, HDL cholesterol, cystatin C and HbA1c were not associated with pneumonia. Low FEV1 was more strongly associated with pneumonia than with COVID-19. Pneumonia also showed an association with baseline cancer.

### Population attributable risks

Factors that were significant in table 2 were mutually adjusted to compare their PAFs for COVID-19 and pneumonia (table 3). Among potentially modifiable risk factors smoking accounted for 14.9% of COVID-19 cases that occurred within the UK Biobank population, obesity with 6.3%, high HbA1C with 5.3%, treated BP with 5.1%, and slow walking pace with 4.0% (total 35.6%). In contrast, none of these factors were large contributors to pneumonia cases within UK Biobank (table 3).

### DISCUSSION

In this population-based study, we found that confirmed COVID-19 infection was associated with a number of modifiable risk factors, a trend that less apparent for pneumonia and influenza. In particular, the associations of smoking, BMI (and body fat), hypertension and physical fitness (as measured by slow walking pace) with COVID-19 are of note; even when such factors were measured a decade before infection they potentially accounted for one-third of COVID-19 cases in UK Biobank. The associations of FEV1 and BMI with COVID-19 are linear, suggesting even modest improvements in lifestyle may be beneficial to risk of presumed severe COVID-19 symptoms. The other independent risk factors for COVID-19 infection included older age, male sex, black ethnicity, socioeconomic deprivation, longstanding illness and reduced renal function as measured by cystatin C, the latter also notable given renal complication in severe COVID-19. Of note, the modifiable risk factors explained some of the association of age with COVID-19 in our adjusted model.

Since this analysis was conducted,[15] several other studies have also shown the association between modifiable and non-modifiable risk factors and COVID-19 risk,[22–29] particularly with obesity. Generally, these data support our analyses showing the importance of both modifiable and non-modifiable risk factors. It is important to recognise the competing risk for health of lockdown and social distancing. Social distancing reduces viral transmission, but also has consequences for lifestyle. Previous

**Table 2** Association of risk factors for COVID-19 and pneumonia in UK Biobank

| | In-hospital COVID-19 | | | | Pneumonia (after 2016) | | | |
| | Model 1 | | Model 2 | | Model 1 | | Model 2 | |
| | RR (95% CI) | P value | RR (95% CI) | P value | RR (95% CI) | P value | RR (95% CI) | P value |
|---|---|---|---|---|---|---|---|---|
| Age (per 5 years) | 1.12 (1.05 to 1.19) | 0.0007 | 1.04 (0.96 to 1.11) | 0.32 | 1.21 (1.11 to 1.33) | <0.0001 | 1.12 (1.01 to 1.24) | 0.03 |
| Male sex | 1.36 (1.11 to 1.66) | 0.003 | 1.72 (1.25 to 2.35) | 0.0008 | 0.58 (0.44 to 0.77) | 0.0002 | 0.80 (0.56 to 1.16) | 0.24 |
| Ethnicity | | | | | | | | |
| White | 1 (Reference) | – | 1 (Reference) | – | 1 (Reference) | – | 1 (Reference) | – |
| Black | 2.32 (1.33 to 4.04) | 0.003 | 2.00 (1.13 to 3.53) | 0.02 | 1.63 (0.51 to 5.21) | 0.41 | 1.20 (0.37 to 3.89) | 0.76 |
| South Asian | 1.98 (1.10 to 3.55) | 0.02 | 1.73 (0.94 to 3.17) | 0.08 | 0.96 (0.31 to 3.05) | 0.95 | 0.66 (0.20 to 2.14) | 0.49 |
| Others | 1.83 (1.08 to 3.10) | 0.03 | 1.70 (0.99 to 2.91) | 0.053 | 0.63 (0.15 to 2.55) | 0.51 | 0.52 (0.13 to 2.15) | 0.37 |
| Deprivation index (per 1 SD) | 1.27 (1.16 to 1.39) | <0.0001 | 1.17 (1.06 to 1.29) | 0.001 | 1.10 (0.96 to 1.26) | 0.18 | 1.04 (0.90 to 1.20) | 0.62 |
| Current/former smoker | 1.45 (1.19 to 1.79) | 0.0003 | 1.39 (1.13 to 1.71) | 0.002 | 0.95 (0.72 to 1.25) | 0.71 | 0.89 (0.67 to 1.19) | 0.44 |
| Alcohol drinking status | | | | | | | | |
| Never | 1 (Reference) | – | – | – | 1 (Reference) | – | – | – |
| Former | 0.90 (0.49 to 1.65) | 0.74 | – | – | 1.28 (0.58 to 2.79) | 0.54 | – | – |
| Current | 0.65 (0.43 to 1.00) | 0.05 | – | – | 0.66 (0.37 to 1.18) | 0.16 | – | – |
| BMI (per 1 SD) | 1.36 (1.25 to 1.48) | <0.0001 | 1.28 (1.16 to 1.40) | <0.0001 | 1.25 (1.11 to 1.41) | 0.0002 | 1.16 (1.02 to 1.32) | 0.03 |
| BMI categories | | | | | | | | |
| Underweight | 1.17 (0.16 to 8.59) | 0.88 | – | – | 1.43 (0.20 to 10.04) | 0.73 | – | – |
| Normal | 1 (Reference) | – | – | – | 1 (Reference) | – | – | – |
| Overweight | 1.43 (1.10 to 1.87) | 0.008 | – | – | 1.01 (0.72 to 1.42) | 0.94 | – | – |
| Obese | 2.08 (1.58 to 2.74) | <0.0001 | – | – | 1.44 (1.01 to 2.05) | 0.04 | – | – |
| Body fat-free mass (per 1 SD) | 1.39 (1.17 to 1.64) | 0.0002 | – | – | 1.42 (1.10 to 1.84) | 0.008 | – | – |
| Body fat mass (per 1 SD) | 1.37 (1.25 to 1.50) | <0.0001 | – | – | 1.27 (1.12 to 1.44) | 0.0002 | – | – |
| Body fat per cent (per 1 SD) | 1.58 (1.37 to 1.81) | <0.0001 | – | – | 1.34 (1.11 to 1.62) | 0.002 | – | – |
| Systolic blood pressure (per 1 SD) | 1.01 (0.91 to 1.12) | 0.89 | – | – | 0.94 (0.81 to 1.08) | 0.39 | – | – |
| Diastolic blood pressure (per 1 SD) | 1.06 (0.96 to 1.17) | 0.25 | – | – | 0.90 (0.79 to 1.04) | 0.16 | – | – |
| FEV1 (per 1 SD) | 0.76 (0.66 to 0.87) | <0.0001 | 0.84 (0.73 to 0.97) | 0.02 | 0.70 (0.58 to 0.86) | 0.0005 | 0.75 (0.61 to 0.92) | 0.006 |
| FVC (per 1 SD) | 0.80 (0.70 to 0.93) | 0.003 | – | – | 0.72 (0.59 to 0.90) | 0.003 | – | – |
| FEV1/FVC (per 1 SD) | 0.89 (0.81 to 0.98) | 0.01 | – | – | 0.84 (0.74 to 0.95) | 0.007 | – | – |
| Walking pace | | | | | | | | |
| Slow | 1.99 (1.48 to 2.68) | <0.0001 | 1.53 (1.12 to 2.08) | 0.007 | 1.91 (1.26 to 2.90) | 0.002 | 1.60 (1.04 to 2.46) | 0.03 |
| Average | 1 (Reference) | – | 1 (Reference) | – | 1 (Reference) | – | 1 (Reference) | – |
| Brisk | 0.80 (0.64 to 1.00) | 0.051 | 0.95 (0.75 to 1.20) | 0.65 | 0.78 (0.57 to 1.06) | 0.11 | 0.88 (0.64 to 1.21) | 0.42 |
| Grip strength (per 1 SD) | 0.87 (0.75 to 1.01) | 0.07 | 0.94 (0.81 to 1.10) | 0.45 | 0.85 (0.68 to 1.07) | 0.16 | 0.96 (0.76 to 1.22) | 0.76 |
| Prevalent disease at baseline | | | | | | | | |
| Longstanding illness | 1.44 (1.17 to 1.77) | 0.0005 | 1.16 (0.93 to 1.44) | 0.19 | 1.65 (1.25 to 2.18) | 0.0004 | 1.40 (1.04 to 1.88) | 0.03 |

Continued

**Table 2** Continued

| | In-hospital COVID-19 | | | | Pneumonia (after 2016) | | | |
| | Model 1 | | Model 2 | | Model 1 | | Model 2 | |
| | RR (95% CI) | P value | RR (95% CI) | P value | RR (95% CI) | P value | RR (95% CI) | P value |
|---|---|---|---|---|---|---|---|---|
| Diabetes | 1.72 (1.22 to 2.43) | 0.002 | 1.24 (0.87 to 1.77) | 0.23 | 2.06 (1.29 to 3.30) | 0.002 | 1.59 (0.98 to 2.59) | 0.06 |
| CVD | 1.56 (1.09 to 2.23) | 0.02 | 1.22 (0.85 to 1.76) | 0.29 | 1.64 (1.00 to 2.70) | 0.052 | 1.35 (0.81 to 2.24) | 0.25 |
| Cancer | 1.22 (0.84 to 1.76) | 0.29 | 1.20 (0.83 to 1.74) | 0.33 | 1.72 (1.14 to 2.60) | 0.009 | 1.72 (1.14 to 2.60) | 0.01 |
| Depression | 0.96 (0.73 to 1.25) | 0.74 | 1.01 (0.77 to 1.32) | 0.94 | 0.81 (0.56 to 1.18) | 0.27 | 0.86 (0.59 to 1.25) | 0.42 |
| Asthma | 1.19 (0.89 to 1.59) | 0.23 | 1.04 (0.78 to 1.40) | 0.78 | 1.44 (0.99 to 2.09) | 0.054 | 1.25 (0.85 to 1.82) | 0.25 |
| COPD | 2.80 (0.89 to 8.82) | 0.08 | 1.81 (0.57 to 5.79) | 0.31 | – | – | – | – |
| Bronchitis | 1.65 (0.82 to 3.36) | 0.16 | 1.26 (0.61 to 2.58) | 0.53 | 2.23 (0.98 to 5.06) | 0.055 | 1.77 (0.77 to 4.07) | 0.18 |
| Pneumonia | 1.57 (0.77 to 3.17) | 0.21 | 1.50 (0.74 to 3.05) | 0.26 | 1.53 (0.57 to 4.14) | 0.40 | 1.47 (0.54 to 3.99) | 0.45 |
| Other respiratory disease | 1.36 (0.43 to 4.26) | 0.60 | 1.21 (0.38 to 3.82) | 0.74 | 0.84 (0.12 to 6.04) | 0.86 | 0.76 (0.10 to 5.52) | 0.78 |
| Medication at baseline | | | | | | | | |
| Statin | 1.53 (1.20 to 1.95) | 0.0006 | 1.24 (0.97 to 1.59) | 0.09 | 1.08 (0.75 to 1.55) | 0.69 | 0.91 (0.62 to 1.32) | 0.62 |
| BP medication | 1.62 (1.28 to 2.06) | <0.0001 | 1.33 (1.04 to 1.70) | 0.02 | 0.96 (0.66 to 1.41) | 0.85 | 0.82 (0.55 to 1.21) | 0.31 |
| Steroid | 0.63 (0.09 to 4.57) | 0.65 | 0.53 (0.07 to 3.83) | 0.53 | 1.20 (0.17 to 8.63) | 0.86 | 1.03 (0.14 to 7.53) | 0.97 |
| Biomarker | | | | | | | | |
| Total cholesterol (per 1 SD) | 0.89 (0.81 to 0.99) | 0.03 | 0.93 (0.85 to 1.03) | 0.16 | 0.94 (0.82 to 1.08) | 0.40 | 0.98 (0.85 to 1.12) | 0.76 |
| HDL cholesterol (per 1 SD) | 0.73 (0.65 to 0.83) | <0.0001 | 0.83 (0.73 to 0.95) | 0.005 | 0.88 (0.76 to 1.03) | 0.11 | 0.98 (0.84 to 1.16) | 0.84 |
| Cystatin C (per 1 SD) | 1.27 (1.16 to 1.39) | <0.0001 | 1.13 (1.02 to 1.25) | 0.02 | 1.08 (0.94 to 1.24) | 0.29 | 0.96 (0.83 to 1.12) | 0.59 |
| HbA1c (per 1 SD) | 1.23 (1.15 to 1.32) | <0.0001 | 1.14 (1.05 to 1.23) | 0.001 | 1.19 (1.07 to 1.33) | 0.001 | 1.11 (0.99 to 1.25) | 0.07 |
| CRP (per 1 SD) | 1.14 (1.06 to 1.22) | 0.0006 | 1.03 (0.95 to 1.13) | 0.45 | 1.11 (1.00 to 1.24) | 0.04 | 1.03 (0.91 to 1.16) | 0.67 |
| Rheumatoid factor (per 1 SD) | 1.02 (0.93 to 1.12) | 0.70 | 1.01 (0.92 to 1.11) | 0.82 | 1.03 (0.91 to 1.15) | 0.68 | 1.02 (0.90 to 1.15) | 0.76 |
| Red cell distribution width (per 1 SD) | 1.08 (0.99 to 1.19) | 0.09 | 1.03 (0.94 to 1.13) | 0.55 | 1.12 (0.99 to 1.27) | 0.07 | 1.07 (0.95 to 1.22) | 0.27 |
| White cell count (per 1 SD) | 1.15 (1.05 to 1.27) | 0.003 | 1.04 (0.94 to 1.15) | 0.44 | 1.15 (1.01 to 1.32) | 0.03 | 1.07 (0.93 to 1.23) | 0.32 |
| Neutrophil count (per 1 SD) | 1.12 (1.02 to 1.23) | 0.02 | 1.03 (0.93 to 1.13) | 0.61 | 1.11 (0.97 to 1.27) | 0.12 | 1.04 (0.91 to 1.19) | 0.57 |
| Lymphocyte count (per 1 SD) | 1.11 (1.01 to 1.23) | 0.02 | 1.03 (0.94 to 1.14) | 0.53 | 1.07 (0.94 to 1.23) | 0.28 | 1.02 (0.89 to 1.17) | 0.76 |
| Monocyte count (per 1 SD) | 1.09 (0.99 to 1.20) | 0.08 | 1.01 (0.92 to 1.12) | 0.79 | 1.11 (0.97 to 1.28) | 0.12 | 1.06 (0.92 to 1.21) | 0.43 |

Data presented as risk ratios and their 95% CI. Continuous exposures were standardised and presented as 1 SD increment. Variables with associations denoted "-" are excluded from the model due to non-significance in previous model and / or collinearity with another variable. Analyses were adjusted for model 1 (age, sex, ethnicity and deprivation) and model 2 (model 1 plus body mass index (BMI), forced expiratory volume in 1 s (FEV1), and walking pace (and also grip strength and diastolic blood pressure for COVID-19 only))

BP, blood pressure; COPD, chronic obstructive pulmonary disease; CRP, C reactive protein; CVD, cardiovascular disease; FVC, forced vital capacity.

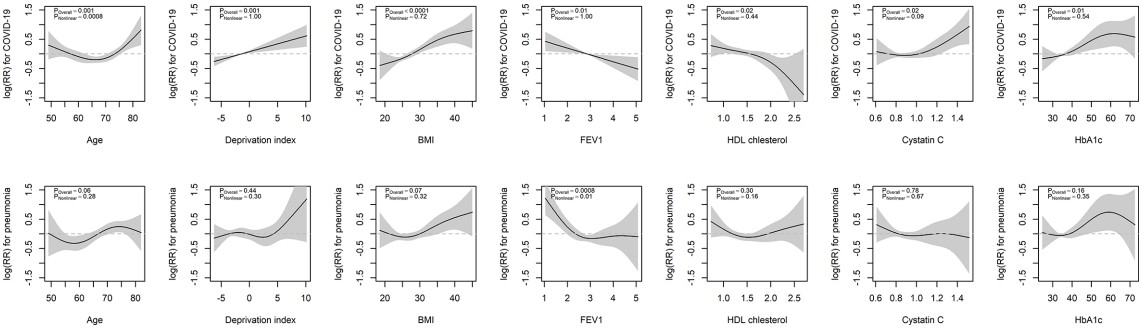

**Figure 1** Non-linear associations of significant continuous variables with COVID-19 and pneumonia. BMI, bodty mass index; FEV1, forced expiratory volume in one second.

data suggest that physical fitness can be rapidly lost when activity levels decrease,[30 31] and this will also result in an increase in BMI.[32 33] Further, social distancing may increase loneliness, depression and psychological stress in some people and consequently adversely affect eating habits[34] and other health behaviours. Anecdotal evidence from our clinics and media suggest many are struggling with overeating. This study suggests public health guidance should focus on reducing the risk of severe complications of COVID-19 by advocating a healthy lifestyle during the ongoing pandemic, not just for general cardiovascular and metabolic health, but also to help to protect against COVID-19 infection, used alongside other public health interventions.

Understanding the actual causes of disease, rather than markers that simply correlate with exposures, is clearly a key issue to consider. The independent association of socioeconomic deprivation with COVID-19 after multiple adjustment may be explained by the accumulation of earlier life socioeconomic adversities which can result in less physiological reserve and more multimorbidity.[34 35] It may also be linked to more overcrowding, reduced social distancing and potential exposure to greater viral load. Asthma, diabetes and high blood pressure, which have been shown to be associated with a higher risk of severe COVID-19 outcomes[36] also showed some trends to be associated with COVID-19 in this dataset. While substantial focus of COVID-19 research has been its apparently

more aggressive symptoms and disease progression in older people, it is important to recognise that people who are older have less cardiorespiratory reserve to cope with COVID-19 infection. Older age is also associated with more hypertension and diabetes, poorer lung function and greater relative fat mass.[34]

Previous reports have suggested that obesity or excess ectopic fat deposition may be a unifying risk factor for severe COVID-19 infection,[8 37] reducing both protective cardiorespiratory reserve as well as potentiating the immune dysregulation that appears, at least in part, to mediate the progression to critical illness in a proportion of patients with COVID-19. Our analysis bears out these hypotheses. Once BMI or body fat was adjusted for, inflammatory markers were no longer associated with incident COVID-19. This suggests that proinflammatory markers, arising from increased adipose deposition, are probably acting as a marker for body fat which seems to be an adverse risk factor for severe COVID-19.[8 37–39] Furthermore, obesity enhances thrombosis,[40] which is relevant given the association between severe COVID-19 and prothrombotic disseminated intravascular coagulation and high rates of venous thromboembolism,[41 42] as well as the association with D-dimer seen in other reports.[43] D-dimer was not measured in blood samples in UK Biobank.

Given that pneumonia is a critical clinical complication of COVID-19, it is important to understand how risk

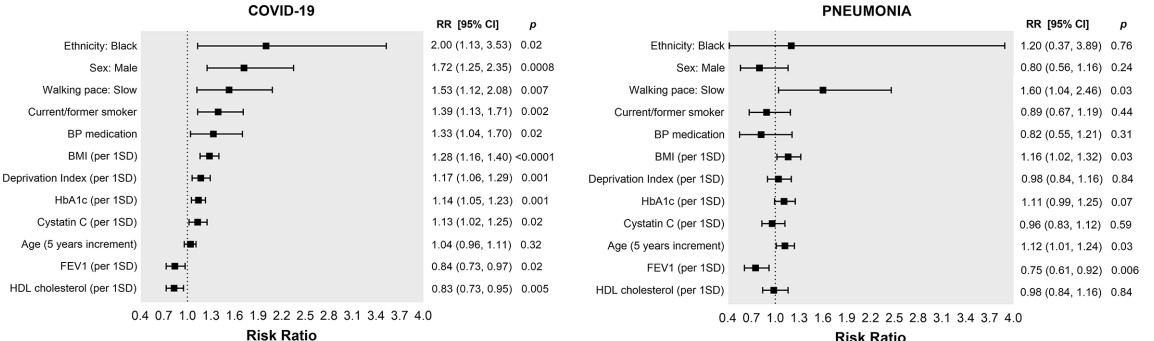

**Figure 2** Risk factors for COVID-19 and pneumonia in the UK Biobank cohort. Data presented as risk ratios (RR) and their 95% CI. Analyses were adjusted for age, sex, ethnicity, deprivation, body mass index (BMI), forced expiratory volume in 1 s (FEV1) and walking pace (and diastolic blood pressure for COVID-19 only). Continuous exposures were standardised and presented per 1-SD increment (deprivation index SD=3.01, cystatin C SD=0.14, BMI SD=4.59, HbA1c SD=5.80, FEV1 SD=0.77 and HDL cholesterol SD=0.37). BP, blood pressure.

**Table 3** Population attributable fractions of COVID-19 in the UK Biobank

| | COVID-19 | | Pneumonia after 2016 | |
|---|---|---|---|---|
| | RR (95% CI) | PAF (%) | RR (95% CI) | PAF (%) |
| Current/former smoker | 1.39 (1.13 to 1.70) | 14.94 | 0.91 (0.69 to 1.20) | −4.15 |
| High deprivation index | 1.45 (1.16 to 1.82) | 8.33 | 1.00 (0.71 to 1.40) | −0.10 |
| Male | 1.18 (0.94 to 1.50) | 8.00 | 0.64 (0.46 to 0.90) | −20.65 |
| Age >65 years | 1.18 (0.94 to 1.46) | 6.33 | 1.29 (0.96 to 1.73) | 9.88 |
| BMI>30 kg/m$^2$ | 1.29 (1.03 to 1.61) | 6.25 | 1.14 (0.83 to 1.57) | 3.17 |
| High HbA1c | 1.29 (1.02 to 1.62) | 5.26 | 1.37 (1.00 to 1.88) | 6.66 |
| Low HDL cholesterol | 1.27 (1.00 to 1.61) | 5.15 | 1.34 (0.95 to 1.91) | 6.45 |
| BP medication | 1.34 (1.05 to 1.71) | 5.11 | 0.88 (0.60 to 1.29) | −2.19 |
| High cystatin C | 1.24 (0.98 to 1.56) | 4.48 | 1.12 (0.80 to 1.57) | 2.39 |
| Slow walking pace | 1.66 (1.23 to 2.24) | 4.02 | 1.78 (1.17 to 2.70) | 4.69 |
| Ethnicity: black | 1.87 (1.32 to 2.65) | 4.00 | 0.70 (0.34 to 1.45) | −2.00 |
| Low FEV1 | 1.15 (0.88 to 1.50) | 2.93 | 1.64 (1.18 to 2.28) | 11.23 |

Estimated using prevalence in the study sample and RR shown in this table.
All factors shown are mutually adjusted. Continuous variables were categorised with the top 20% as the at-risk group.
RR shown in this table may be over adjusted, for example, cystatin C and HDL cholesterol may be downstream factors of body mass index (BMI).
BP, blood pressure; FEV1, forced expiratory volume in 1 s.

factors for COVID-19 related pneumonia differ from 'classic' community acquired pneumonia.[11] Our comparison between other common respiratory diseases and COVID-19 suggests that identification of at-risk groups based on our understanding of other respiratory diseases may be inadequate—a more refined approach to risk stratification based on COVID-19 specific risks is needed, and future data should focus on some of the exposures we identify to achieve this.

The strengths of our study include the large cohort size at an age relevant to more severe COVID-19 symptoms, and biochemistry assays performed in a single dedicated central laboratory. We were also able to extensively explore emerging and novel risk factors for COVID-19, while simultaneously comparing to risk factors for pneumonia identified to primary care providers. Limitations include that UK Biobank is not representative of the whole UK population[44] (our data focus on England specifically) although this is generally not a concern in investigating risk associations.[45] Care should be taken in generalising the PAF estimates. These related to cases occurring within the UK Biobank population, but are not directly applicable to the general population where the prevalence of risk factors is different. In addition, mutual adjustment for overlapping risk factors can lead to problems in interpretability of the PAFs.[46] However, the direction of causal associations with this new outcome is not clear, and these exploratory analyses allow a direct comparison between COVID-19 and pneumonia in the same cohort. Due to the underrepresentation of non-white ethnicities in UK Biobank, we have limited power to explore important interactions by ethnic group (although black ethnicity was associated

with the outcome), and we recognise this as an important risk factor. Ascertainment bias, including differential healthcare seeking, differential testing and differential prognosis may explain some differences in outcomes given poor coverage of testing in the UK. It is also likely that cases will generally be at the more severe end of the clinical spectrum by using hospitalised cases as the outcome, although use of admission to Intensive Care units would be additionally informative once sufficient case numbers accrue. Despite this, we still observe many similar risk factors to incident pneumonia, which is more likely to have close to complete case ascertainment in primary care records, although, as discussed, other risk factors seem more strongly linked to COVID-19 infections. Exposures were measured several years before the development of the outcomes, and misclassification of the exposures will likely bias our results to the null. However, this also serves to illustrate that the risk factors were generally present many years before development of the disease, and as is well known, risk factors tend to track with age. We have also been unable to fully exclude all deaths that occurred prior to the pandemic, due to lack of up-to-date linkage to mortality records.

In conclusion, these data from UK Biobank suggest risk factors for confirmed COVID-19 infection differ in some important ways from risk factors for pneumonia, being more common in men than women, in lower SES, and with stronger associations with ethnicity, CV risk markers, prior smoking and adiposity. Such findings suggest possible merit in advocating improvements in lifestyle as an additional measure to reduce the risk of COVID-19 alongside existing public health measures such as social distancing and shielding of high risk groups. They also

have implications for health advice targeted at the public to lessen risks during this pandemic.

**Contributors** PW, JPP and NS conceived the idea for the paper. FH conducted the analysis. All authors contributed to the interpretation of the findings. PW, FH, CC-M and NS jointly wrote the first draft. All authors critically revised the paper for intellectual content and approved the final version of the manuscript. PW and NS are guarantors of the work.

**Funding** The work in this study is supported by the British Heart Foundation Centre of Research Excellence Grant RE/18/6/34217. CLN acknowledges funding from a Medical Research Council Fellowship (MR/R024774/1). SVK acknowledges funding from the Medical Research Council (MC_UU_12017/13), Scottish Government Chief Scientist Office (SPHSU13), and NRS Senior Clinical Fellowship (SCAF/15/02).

**Disclaimer** The funders played no part in the research.

**Competing interests** PW has received research grants from Roche Diagnostics, AstraZeneca and Boehringer Ingelheim outside the submitted work, and NS has received grant and personal fees from Boehringer Ingelheim, and personal fees from Amgen, AstraZeneca, Eli Lilly, Novo Nordisk, Pfizer, and Sanofi outside the submitted work. All authors declare no other relationships or activities that could appear to have influenced the submitted work.

**Patient consent for publication** Not required.

**Ethics approval** UK Biobank received ethical approval from the North West Multi-centre Research Ethics Committee (REC reference: 11/NW/03820). All participants gave written informed consent before enrolment in the study, which was conducted in accordance with the principles of the Declaration of Helsinki. Direct dissemination of the results to participants is not possible/applicable. This study was performed under UK Biobank application number 7155.

**Provenance and peer review** Not commissioned; externally peer reviewed.

**Data availability statement** Data may be obtained from a third party and are not publicly available. UK Biobank data can be requested by bona fide researchers for approved projects, including replication, through https://www.ukbiobank.ac.uk/

**ORCID iDs**
Frederick K Ho http://orcid.org/0000-0001-7190-9025
Stuart R Gray http://orcid.org/0000-0001-8969-9636
S Vittal Katikireddi http://orcid.org/0000-0001-6593-9092
Claire L Niedzwiedz http://orcid.org/0000-0001-6133-4168
Naveed Sattar http://orcid.org/0000-0002-1604-2593

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
