## [Reviewer comments · BMJ Open]

ARTICLE DETAILS

TITLE (PROVISIONAL)	Modifiable and non-modifiable risk factors for COVID-19, and comparison to risk factors for influenza and pneumonia: results from a UK Biobank prospective cohort study
AUTHORS	Ho, Frederick; Celis-Morales, Carlos; Gray, Stuart; Katikireddi, Srinivasa; Niedzwiedz, Claire; Hastie, Claire; Ferguson, Lyn; Berry, Colin; Mackay, Daniel; Gill, Jason; Pell, J. P.; Sattar, Naveed; Welsh, Paul

VERSION 1 – REVIEW

REVIEWER	Dr Colin Moran (and Kirstin MacGregor) University of Stirling, Scotland Currently working on a different project with one of the authors (Dr Celis-Morales).
REVIEW RETURNED	17-Jul-2020

GENERAL COMMENTS	The manuscript investigates modifiable and non-modifiable risk factors for COVID-19 and provides a comparison with risk factors of influenza and pneumonia in the UKBB. This has not previously been addressed in a general population. Findings show that modifiable risk factors for COVID-19 include body mass index, forced expiratory volume in 1 second, smoking and physical fitness. Non-modifiable risk factors include biological sex, ethnicity and socioeconomic deprivation. Importantly, modifiable risk factors that can be improved with lifestyle modifications are highlighted. Given the current global COVID-19 pandemic this work adds important information that may be useful when framing public health messages. ----- Minor comments: Title 1. This could include reference to comparing COVID-19 risk factors with risk factors for influenza and pneumonia. This is an important aspect of the manuscript. Abstract 1. COVID is not mentioned in the objectives when it is in fact the focus of the paper. 2. The modifiable risk factors FEV1 and HDL could be made clearer. Lower FEV1 is followed by per SD increase in brackets. No higher/lower stated in text with HDL. Consider flipping risk round to be reported per SD reduction or grouping them without changing as protective higher FEV1 and higher HDL. Methods
---

1. The sentence "To reduce bias, we excluded from the study all participants known to have died COVID-19 pandemic" is not clear. I think there are words missing after died:
...before/during/since/because of the COVID-19...
2. Please also briefly elaborate on how this reduces bias.
3. p7, line 40 define CKD and COPD on first use. In fact, several abbreviations used throughout the manuscript are not defined prior to abbreviation. It would enhance clarity for readers not from a biomedical background to define these terms in text on first use.
4. p7, line 34 add (CVD) following cardiovascular disease.
5. You state that some conditions did not have sufficient numbers of cases for multivariate analysis. Please state what cut off you considered to be sufficient.
6. UKBB holds data for numerous physical activity measures (e.g. past week physical activity, physical activity in the past 4 weeks and typical duration of physical activity). Physical activity is a well-established modifiable risk factor for various diseases, but the relationship between physical activity and risk factors for COVID-19 is unknown. Given the nature of physical activity as a modifiable lifestyle factor, why did you instead use walking speed as a proxy for fitness?
7. Clarity should be improved on the step when you remove variables from the analysis because of their colinearity with other variables. Currently, this is brought in only in model 2 and is a little confusing in presentation. Why only model 2? How did you choose what made them similar factors? Biology or statistics?

Results

1. Similar to comment 1 in the abstract.
2. Inconsistencies in the description of data in the results section reduce clarity of the data. Risk ratios are presented for data in the section "Univariate risk factors for incident COVID-19". It would be useful for consistency and data interpretation to also present risk factors for relevant data in the sections "Univariable risk factors for incident pneumonia" and "Univariable risk factors for incident influenza".
3. Additionally, including 95%CI with RR through the text and tables would aid interpretation.
4. p11, line 22 is inconsistent with results tables. This section implies that corrections were only done for sociodemographics whilst tables (and methods) state that model 1 included a correction for sex, ethnicity, sociodemographics and age.
5. p12, line 9 similar to what? 50-60 yo?
6. Some language could be tightened up to avoid subjective terms such as "more moderately", "more strongly" and "weakly".

Discussion

1. p13/14 last/first sentence. Completely agree. Did you try to correct your models for these factors and see if the relationship with age disappears?

Table 1

1. Confidence intervals on the RR scores in this table and throughout the manuscript would also be useful.
2. The REF group is wrongly allocated for alcohol drinking status. Please check this and amend across row and table.
3. I see the note about "†Evidence for non-linearity"; however, I cannot see the symbol used anywhere in the table. Its a complicated table so I may be missing it but please check as I think it is important to include for the relevant variable - probably in

	the first column although relationship likely different for different conditions. This may need further clarification. Table 2 1. p25, line 24 typo standard deviation. Table 3 1. The mutually adjusted approach may give misleading results. See https://dx.doi.org/10.1186%2Fs12889-018-6364-y for discussion. Please comment on robustness of your analysis to this problem. Figure 1 No comment Figure 2 1. Why is white cell count SD included in the legend? 2. This figure would be easier to interpret if the risk factors were in the same order (y-axis) on both graphs. Supplementary Table 1 1. I think you have the REF group wrongly allocated in ethnicity. Whites are not 4X more likely to suffer from COVID than Blacks. Please check this and amend across row and table. 2. Confidence intervals on the RR scores in this table would also be useful. 3. I think HbA1c RR is inverted. A higher score puts you at more risk not less. 4. Please check Rheumatoid factor as the values are identical across all columns. This seems unlikely given the RR scores. 5. This does not match the description of model 2 in the methods of the main paper. 6. I see the note about "†Evidence for non-linearity"; however, I cannot see the symbol used anywhere in the table. Its a complicate table so I may be missing it but please check as I think it is important to include for the relevant variable - probably in the first column although relationship likely different for different conditions. This may need further clarification. Supplementary Table 2 1. I think REF group is wrongly allocated to former drinker here. Please check this and the rest of the table. 2. Does model 2 here really include BF%? STable 4 suggests that this one includes BMI and BF% is only introduced in STable 4. Supplementary Table 4 1. Only one variable (COPD) has an association denoted "-", was it excluded due to non-significance or collinearity?
--	--

REVIEWER	Tsvetoslav Georgiev Medical University - Varna
REVIEW RETURNED	11-Aug-2020

GENERAL COMMENTS	The manuscript presents modifiable and non-modifiable risk factors for COVID-19. It is a much-needed study and makes a perfect sense in the crisis of COVID-19; the sample size is large enough and the studied risk factors are quite a lot. My comments are more or less cosmetic:
--

	1. The methods section should be restructured including Patients subsection and exposures separation into modifiable and non-modifiable. 2. The statement is unclear and should be fixed: "To reduce bias, we excluded from the study all participants known to have died COVID-19 pandemic." 3. The used tests for the diagnosis should be mentioned - probably PCR - be more specific for the establishment of COVID-19 cases.
--	---

VERSION 1 – AUTHOR RESPONSE

Reviewer: 1

Reviewer Name: Dr Colin Moran (and Kirstin MacGregor) Institution and Country: University of Stirling, Scotland Please state any competing interests or state 'None declared': Currently working on a different project with one of the authors (Dr Celis-Morales).

Please leave your comments for the authors below The manuscript investigates modifiable and non-modifiable risk factors for COVID-19 and provides a comparison with risk factors of influenza and pneumonia in the UKBB. This has not previously been addressed in a general population. Findings show that modifiable risk factors for COVID-19 include body mass index, forced expiratory volume in 1 second, smoking and physical fitness. Non-modifiable risk factors include biological sex, ethnicity and socioeconomic deprivation. Importantly, modifiable risk factors that can be improved with lifestyle modifications are highlighted. Given the current global COVID-19 pandemic this work adds important information that may be useful when framing public health messages.

Minor comments:

Title

1. This could include reference to comparing COVID-19 risk factors with risk factors for influenza and pneumonia. This is an important aspect of the manuscript.

Response: We agree. We have now done this.

Abstract

1. COVID is not mentioned in the objectives when it is in fact the focus of the paper.

Response: We agree this is an oversight. We have now added this.

"We aimed to investigate demographic, lifestyle, socioeconomic, and clinical risk factors for COVID-19, and compared them to risk factors for pneumonia and influenza in UK Biobank."

2. The modifiable risk factors FEV1 and HDL could be made clearer. Lower FEV1 is followed by per SD increase in brackets. No higher/lower stated in text with HDL. Consider flipping risk round to be reported per SD reduction or grouping them without changing as protective higher FEV1 and higher HDL.

Response: We agree. We have now done this.

"Higher FEV1 and HDL-cholesterol were both associated with lower risk (RR 0.84 and RR 0.83 per SD increase respectively)."

Methods

1. The sentence "To reduce bias, we excluded from the study all participants known to have died COVID-19 pandemic" is not clear. I think there are words missing after died:

...before/during/since/because of the COVID-19...

2. Please also briefly elaborate on how this reduces bias.

Response: We agree. We now state

"To reduce competing risks, whereby risk factors may influence risk of death before the pandemic occurred, we excluded from the study all participants known to have died prior to the COVID-19 pandemic."

3. p7, line 40 define CKD and COPD on first use. In fact, several abbreviations used throughout the manuscript are not defined prior to abbreviation. It would enhance clarity for readers not from a biomedical background to define these terms in text on first use.

4. p7, line 34 add (CVD) following cardiovascular disease.

Response: We agree. We have updated definitions throughout and have added “CVD” in the introduction.

5. You state that some conditions did not have sufficient numbers of cases for multivariate analysis. Please state what cut off you considered to be sufficient.

Response: This is somewhat a judgement call, since whether the data were included in multivariable analysis also depends on issues like collinearity. We have therefore rephrased “Some conditions were not included in multivariable analysis due to low case numbers and/or high correlations with other conditions, but univariable results are presented for completeness.”

6. UKBB holds data for numerous physical activity measures (e.g. past week physical activity, physical activity in the past 4 weeks and typical duration of physical activity). Physical activity is a well-established modifiable risk factor for various diseases, but the relationship between physical activity and risk factors for COVID-19 is unknown. Given the nature of physical activity as a modifiable lifestyle factor, why did you instead use walking speed as a proxy for fitness?

Response: We agree this is somewhat a subjective measure, but our other UKB publications (and others) have repeatedly found that walking pace is one of the best biomarkers for poor health outcomes in general – regardless of whether it is truly a marker reflective of nuanced physical activity in the more academic sense. In addition, walking pace is better correlated with cardiorespiratory fitness than physical activity in the UK Biobank, which may be explained by the self-reported nature and dilution bias related to the physical activity questionnaire.

<https://pubmed.ncbi.nlm.nih.gov/32299669/>

<https://pubmed.ncbi.nlm.nih.gov/31168053/>

<https://pubmed.ncbi.nlm.nih.gov/29020281/>

We also report more objective markers in grip strength, FEV1 and FVC, and these are still included in table 3.

7. Clarity should be improved on the step when you remove variables from the analysis because of their colinearity with other variables. Currently, this is brought in only in model 2 and is a little confusing in presentation. Why only model 2? How did you choose what made them similar factors? Biology or statistics?

Response: As suggested, we have elaborated the two reasons behind selecting one out of a group of variables. The first reason is because some variables were different operationalisation of the same concept (e.g. BMI and BMI categories) and the second is high correlation between conceptually similar variables (e.g. $r=0.87$ between BMI, and body fat mass, and $r=0.97$ between FEV1 and FVC). Variables in model 1 do not have to contend with the above two issues.

Results

1. Similar to comment 1 in the abstract.

Response: We agree this is an oversight. We have now added this.

2. Inconsistencies in the description of data in the results section reduce clarity of the data. Risk ratios are presented for data in the section “Univariate risk factors for incident COVID-19”. It would be useful for consistency and data interpretation to also present risk factors for relevant data in the sections “Univariable risk factors for incident pneumonia” and “Univariable risk factors for incident influenza”.

Response: We agree, and have updated these sections as you suggest.

3. Additionally, including 95%CI with RR through the text and tables would aid interpretation.

Response: Since the tables are large and these are only univariable analyses we have opted to present only point estimates and p-values here, while presenting 95%CI for more important multivariable models.

4. p11, line 22 is inconsistent with results tables. This section implies that corrections were only done for sociodemographics whilst tables (and methods) state that model 1 included a correction for sex, ethnicity, sociodemographics and age.

Response: We have clarified

“After multivariable adjustment for age, sex, ethnicity, and socioeconomic status”

5. p12, line 9 similar to what? 50-60 yo?

Response: We have clarified

“Age was associated with COVID-19 admission in a J-shaped curve, where for participants aged 60-70 years the curve was relatively flat, and for participants aged over 75 years risk increased exponentially with age”

6. Some language could be tightened up to avoid subjective terms such as “more moderately”, “more strongly” and “weakly”.

Response: We agree and have removed some of this language but have retained some of these words where they pertain to direct comparison of risk factors for the different conditions.

Discussion

1. p13/14 last/first sentence. Completely agree. Did you try to correct your models for these factors and see if the relationship with age disappears?

Response: We agree this is an important point and we have clarified.

“Of note, the modifiable risk factors explained some of the association of age with COVID-19 in our adjusted model.”

Table 1

1. Confidence intervals on the RR scores in this table and throughout the manuscript would also be useful.

Response: Since the tables are large and these are only univariable analyses we have opted to present only point estimates and p-values here, while presenting 95%CI for more important multivariable models.

2. The REF group is wrongly allocated for alcohol drinking status. Please check this and amend across row and table.

Response: Apologies for the typo. The reference group for alcohol drinking should be never drinker. This is now corrected.

3. I see the note about “†Evidence for non-linearity”; however, I cannot see the symbol used anywhere in the table. Its a complicated table so I may be missing it but please check as I think it is important to include for the relevant variable - probably in the first column although relationship likely different for different conditions. This may need further clarification.

Response: Thank you, we agree this is important. Corrected and clarified reference to supplementary table 3.

Table 2

1. p25, line 24 typo standard deviation.

Response: Thank you, corrected.

Table 3

1. The mutually adjusted approach may give misleading results. See <https://dx.doi.org/10.1186%2Fs12889-018-6364-y> for discussion. Please comment on robustness of your analysis to this problem.

Response: We agree that this is an issue worthy of discussion, and we have debated it internally. The issue is that we are uncertain (for example) we are uncertain whether obesity causes severe COVID-19, or whether poor lung function exacerbated by BMI is the causal agent. There are many ways to infer causal associations here, and nothing is clear-cut. Our analysis, which was one of the first large epidemiological studies of COVID-19 in the general population (<https://www.medrxiv.org/content/10.1101/2020.04.28.20083295v1.full.pdf>), is intended to highlight the fact that general healthy lifestyle seems to be important, more than individual risk factors. Essentially, we have added a comment to the limitations section and referenced the paper “In addition, mutual adjustment for overlapping risk factors can lead to problems in interpretability of the PAFs. However, the direction of causal associations with this new outcome is not clear, and these exploratory analyses allow a direct comparison between COVID-19 and pneumonia in the same cohort.”

Figure 1

No comment

Figure 2

1. Why is white cell count SD included in the legend?

Response: Thank you, we have removed this

2. This figure would be easier to interpret if the risk factors were in the same order (y-axis) on both graphs.

Response: Thank you, we agree and have changed this.

Supplementary Table 1

1. I think you have the REF group wrongly allocated in ethnicity. Whites are not 4X more likely to suffer from COVID than Blacks. Please check this and amend across row and table.

Response: Thank you for pointing this out. The RRs for Whites and Blacks were indeed transposed. This is now corrected.

2. Confidence intervals on the RR scores in this table would also be useful.

Response: Since the tables are large and these are only univariable analyses we have opted to present only point estimates and p-values here, while presenting 95%CI for more important multivariable models.

3. I think HbA1c RR is inverted. A higher score puts you at more risk not less.

Response: Apologies. There were now corrected as well.

4. Please check Rheumatoid factor as the values are identical across all columns. This seems unlikely given the RR scores.

Response: We have clarified

"Biomarkers with data below the limit of detection were imputed as the square root of the limit of detection. The majority of participants had undetectable rheumatoid factor and risk ratios were derived for detectable rheumatoid factor, with the referent being undetectable."

5. This does not match the description of model 2 in the methods of the main paper.

Response: To clarify: this table only represents univariable analysis and was not adjusted for any other factors in regression models.

6. I see the note about "†Evidence for non-linearity"; however, I cannot see the symbol used anywhere in the table. Its a complicate table so I may be missing it but please check as I think it is important to include for the relevant variable - probably in the first column although relationship likely different for different conditions. This may need further clarification.

Response: We have removed this reference from this table. Table 1 (main document) now refers to table s3 for non-linearity.

Supplementary Table 2

1. I think REF group is wrongly allocated to former drinker here. Please check this and the rest of the table.

Response: Thank you. This is now corrected.

2. Does model 2 here really include BF%? STable 4 suggests that this one includes BMI and BF% is only introduced in STable 4.

Response: Indeed, BMI was adjusted, instead of BF%. This is now corrected.

Supplementary Table 4

1. Only one variable (COPD) has an association denoted "-", was it excluded due to non-significance or collinearity?

Response: There were no COPD patients who were diagnosed with pneumonia after 2016. This has been clarified.

Reviewer: 2

Reviewer Name: Tsvetoslav Georgiev

Institution and Country: Medical University - Varna Please state any competing interests or state 'None declared': None declared.

Please leave your comments for the authors below The manuscript presents modifiable and non-modifiable risk factors for COVID-19. It is a much-needed study and makes a perfect sense in the crisis of COVID-19; the sample size is large enough and the studied risk factors are quite a lot. My comments are more or less cosmetic:

1. The methods section should be restructured including Patients subsection and exposures separation into modifiable and non-modifiable.

Response: We have added a patient subsection, and clarified in the methods what we consider the modifiable risk factors to be, without changing the structure substantially. We hope you agree this improves clarity.

““Modifiable” risk factors were considered to include smoking, anthropometric measurements, glycaemic control (HbA1C), lung function measurements, hypertension, HDL-cholesterol, and measures of physical activity.”

2. The statement is unclear and should be fixed: "To reduce bias, we excluded from the study all participants known to have died COVID-19 pandemic."

Response: We agree. We now state

“To reduce competing risks, whereby risk factors may influence risk of death before the pandemic occurred, we excluded from the study all participants known to have died prior to the COVID-19 pandemic.”

3. The used tests for the diagnosis should be mentioned - probably PCR - be more specific for the establishment of COVID-19 cases.

Response: We agree. We now state these were PCR tests and have updated reference to the source data.

VERSION 2 – REVIEW

REVIEWER	Dr Colin Moran (and Kirstin MacGregor) University of Stirling Currently working on a different project with one of the authors (Dr Celis-Morales).
REVIEW RETURNED	25-Sep-2020

GENERAL COMMENTS	I am happy that all my comments have been addressed. This is an important and well written article.
---

REVIEWER	Tsvetoslav Georgiev Medical University - Varna Bulgaria
REVIEW RETURNED	13-Sep-2020

GENERAL COMMENTS	The manuscript was majorly improved mostly due to Rev. #1 comments. I am still thinking that "Exposures" subsection of the methods should be subdivided into modifiable and non-modifiable making the manuscript more "perspicuous". Nevertheless, this structure is also acceptable.
---